# Antileukemic, Antioxidant, Anti-Inflammatory and Healing Activities Induced by a Polyphenol-Enriched Fraction Extracted from Leaves of *Myrtus communis* L.

**DOI:** 10.3390/nu14235055

**Published:** 2022-11-27

**Authors:** Hamza Mechchate, Carlos Eduardo de Castro Alves, Imane Es-safi, Amal Amaghnouje, Fatima Zahra Jawhari, Regiane Costa de Oliveira, Alice de Freitas Gomes, Raffaele Conte, Gemilson Soares Pontes, Dalila Bousta, Andriy Grafov

**Affiliations:** 1Laboratory of Biotechnology, Environment, Agrifood, and Health, University of Sidi Mohamed Ben Abdellah (USMBA), Fez B.P. 1796, Morocco; 2Post-Graduate Program in Basic and Applied Immunology, Institute of Biological Sciences, Federal University of Amazonas, Manaus 69077-000, AM, Brazil; 3Post-Graduate Program in Hematology, The State University of Amazonas, Foundation of Hematology and Hemotherapy of Amazonas, Manaus 69050-010, AM, Brazil; 4Laboratory of Virology and Immunology, National Institute of Amazonian Research (INPA), Manaus 69067-375, AM, Brazil; 5Research Institute on Terrestrial Ecosystems (IRET)—CNR, Via Pietro Castellino 111, 80131 Naples, Italy; 6Materials Chemistry Division, Department of Chemistry, University of Helsinki, A.I. Virtasen aukio 1, 00560 Helsinki, Finland

**Keywords:** anticancer, HL60, K562, wound healing activity, carrageenan-induced edema test, hemolysis test, beta carotene bleaching test, ferric reducing antioxidant power, total antioxidant capacity, acute toxicity study

## Abstract

Natural products have offered a number of exciting approaches in cancer treatment over the years. In this study, we investigated the prophylactic and therapeutic effects of the polyphenol-enriched fraction extracted from *Myrtus communis* (PEMC) on acute and chronic leukemia. According to the UHPLC-MS^n^, the fraction is rich in flavonoids. Protective activity of the PEMC was assessed by evaluating the antioxidant, anti-inflammatory, wound healing, and hemolysis potential in a series of in vivo and in vitro assays, while the therapeutic approach consisted of the evaluation of cytotoxic activity of the PEMC against HL60 and K562 leukemia cell lines. Safety of the fraction was also evaluated on a non-cancerous Vero cell line and by an acute toxicity test performed in mice. The PEMC demonstrated a significant anti-inflammatory and healing potential. The activities found at the dose of 100 mg/kg were better than those observed using a reference drug. The PEMC demonstrated a significant antioxidant effect and a specific cytotoxicity towards HL60 (IC_50_ = 19.87 µM) and K562 (IC_50_ = 29.64 µM) cell lines being non-toxic to the Vero cell line. No hemolytic activity was observed in vitro and no toxicity effect was found in mice. Thus, the PEMC has a pharmacological potential as both preventive and therapeutic agent. However, further research is necessary to propose its mechanism of action.

## 1. Introduction

Bioactive products prepared from natural sources have a number of biological and pharmacological properties, they have been extensively used as a source of medicine since the ancient times [1,2]. Natural products from plant sources have been attracting a constantly growing interest owing to many different pharmacological activities, such as antioxidant, anticancer, immunomodulatory, anti-inflammatory, antidiabetic, diuretic, osteogenic, wound healing, and so on [3,4]. Furthermore, in recent years, evaluation of naturally produced bioactive compounds or extracts has resulted in identification of many active ingredients with high therapeutic potential [2,5,6,7,8,9,10,11].

Cancer is one of the leading causes of mortality worldwide. The incidence of cancer varies significantly and depends on different variables, such as age, gender, race, environment, social status, and geographic origin [12]. There is a number of medicines for cancer treatment; nevertheless, the higher-than-expected toxicity and resistance underscore the need for novel compounds that may either limit the resistance to current therapies or possess a reduced toxicity [13].

Multiple cellular activities in terms of oxidative stress and inflammation may be involved in the genesis and progression of cancer, including cell proliferation, angiogenesis, migration, metabolic reprogramming, and evasion of controlled cell death [14,15]. NF-B, AP-1, p53, HIF-1, PPAR-, catenin/Wnt, and Nrf2 are transcription factors that may be activated by an oxidative stress [16]. Over 500 genes, including growth factors, inflammatory cytokines, chemokines, cell cycle regulatory molecules, and anti-inflammatory molecules, may be expressed upon activation of those transcription factors [17]. Both the oxidative stress and inflammation are cellular outcomes of a biological defense mechanism that may feed cancer and other pathophysiological manifestations [18].

Leukemia, often known as blood cancer, is a hematological disease that starts in the bone marrow, where hematopoietic stem cells are improperly cloned [19]. Acute myeloid leukemia (AML) is a different kind of leukemia that accounts for a majority of occurrences. Fms-like tyrosine kinase 3 (FLT3) has long been recognized to be expressed in the malignant cells of most AML patients [20]. The FLT3 is a transmembrane receptor that controls hematopoietic progenitor cell development, proliferation, and differentiation through the RAS/RAF/MEK/ERK and PI3K/Akt/mTOR pathways [21]. Sunitinib and midostaurin have been authorized as targeted anticancer medicines. However, owing to frequent FLT3 mutations and an emergence of the drug resistance, targeted AML therapies have not progressed much in recent decades [22].

Philadelphia chromosome and associated chimeric oncoprotein BCR-ABL distinguish a chronic myeloid leukemia (CML). The BCR-ABL protein, which has constitutive tyrosine kinase activity, causes persistent activation of downstream signaling cascades (e.g., PI3K/Akt/mTOR) that lead to uncontrolled clonal growth of leukemia cell [23]. In recent years, tyrosine kinase inhibitors (TKI) that target the BCR-ABL have attracted growing attention. However, responses of some CML patients to the TKI treatment were unsatisfactory, since the developing drug resistance began to pose significant therapeutic difficulties [24]. Therefore, the development of less harmful and more effective medicines for leukemia patients remains a critical challenge.

*Myrtus communis* L. is an aromatic shrub of a Myrtaceae family, widespread around the Mediterranean basin and well-known for its medicinal properties since ancient times [25]. A plethora of the plant’s pharmacological properties have been reported, those include antimicrobial, anti-inflammatory, antidiarrheal, antispasmodic, antidiabetic, vasodilator, antiulcer, antioxidant, anticancer, and anxiolytic activities [26]. Mimica-Dukić et al. [27] described that essential oil of the *M. communis* revealed an antimutagenic effect by reducing the percentage of spontaneous and *tert*-butylhydroperoxide-induced mutagenesis in *Escherichia coli.* Several constituents of the oil like myricetin 3-O-galactoside and myricetin 3-O-rhamnoside demonstrated a similar effect, they reduced the mutagenicity caused by nifuroxazide, aflatoxin-B1, and hydrogen peroxide by modulating the expression of several genes involved in the apoptosis, DNA repair, and oxidative stress [28]. A myrtucommulone isolated from myrtle leaves induced apoptotic death of cancer cells by activating caspases, which resulted in a cleavage of poly-(ADP-ribose)polymerase, a release of nucleosomes, a fragmentation DNA, and a potential loss of mitochondrial membrane [29]. The plant extracts are rich in polyphenols, especially in flavonoids, which could be behind the observed activities.

Flavonoids are the most prevalent kind of plant phenol compounds, which may be found in abundance in many edible plants and their products [30]. Over 15,000 distinct flavonoids have been discovered across the plant kingdom, with at least several hundreds of them found in the edible components [31]. Flavonoids are plant secondary metabolites that are not required for the plant’s development or survival. Instead, flavonoids help plants to defend themselves against pests and herbivores, as well as plant diseases, season and climate, environment, and other local constraints, boosting the plant’s overall survival capacity [32,33,34]. Flavonoids have anti-inflammatory properties through different mechanisms such as inhibition of regulatory enzymes and transcription factors that have an important role in the control of mediators involved in the inflammation [35]. They can modulate gene expression via transcription factors, such as NF-κB, GATA-3, STAT-6, and hence the transcription of proinflammatory genes [35]. Flavonoids may also be used as chemotherapeutic and chemopreventive drugs [36,37]. It is a well-known fact that diets rich in fresh fruits and vegetables, which are abundant in A, C, and E vitamins, carotene, flavonoids, and other components, protect against malignancies such as lung, breast, prostate, and colon cancers [38].

Therapeutic properties of flavonoids are also associated with acute and chronic wound healing [39,40]. Flavonoids act on fibroblasts, endothelial cells, and macrophages by playing a regulatory role in a dynamic balance between pro- and anti-inflammatory responses [41]. Skin lesions are a serious public health problem worldwide. Only in the United States, 6.5 million people suffer from chronic or complex wounds, which cost approximately USD 10 billion annually [42,43]. Thus, a continuous search for therapeutic alternatives to the available treatment is necessary to improve both the health assistance and the treatment cost-effectiveness.

In the present study, we investigated the antileukemic, anti-inflammatory, antioxidant, and healing potential of the polyphenol-enriched fraction extracted from leaves of *M. communis* looking for an effective and safe new potential alternative to conventional drugs.

## 2. Materials and Methods

### 2.1. Materials and Reagents

The following commercial cell lines and reagents were used in the study: HL60 (ATCC^®^ CCL-240TM-Human acute promyelocytic leukemia), K562 (ATCC^®^ CCL-243TM—chronic myelogenous leukemia), and Vero cells (kidney epithetical cell line derived from African green monkey, *Chlorocebus* sp.). RPM 1640 and DMEM—Dulbecco’s Modified Eagle Medium, both from Gibco (Rockville, MD, USA) were used as culture media. Heat inactivated fetal bovine serum with Penicillin, Amphotericin B, and Streptomycin (FBS, Gibco) was used to supplement the media. 3-(4,5-dimethyl thiazol-2-yl)-2, 5-diphenyltetrazolium bromide (MTT, Merck, Darmstadt, Germany) was used in the cell viability assays.

### 2.2. Plant Material

The leaves of *Myrtus communis* L. were harvested in August 2018 in the province of Taounate (34.4913° N, 5.1263° W), Morocco. Their botanical identification was carried out by a specialist botanist, and a voucher specimen (BPRN60) was deposited at the Laboratory of Biotechnology, Environment, Agrifood and Health, Faculty of Sciences Dhar el Mahraz, University of Sidi Mohamed Ben Abdellah, Fez, Morocco.

### 2.3. Polyphenol Extraction

Polyphenols are usually obtained from plant materials by liquid phase extraction. The extraction was performed with 100 mL of aqueous methanol 70% in an ultrasonic apparatus at 35 kHz frequency at 40 °C. The powder of dried leaves of *M. communis* (10 g) was sonicated for 45 min, and the extract was filtered at 40 °C through a filter paper and then concentrated on a rotary evaporator. The product was washed twice with dichloromethane and chloroform to eliminate pigments and to obtain the fraction enriched in polyphenols, which was concentrated until dryness on a rotary evaporator. A solid fraction (0.27 g, yield 2.7%) was stored at 4 °C. It was dissolved in a distilled water before further experiments [44].

### 2.4. Polyphenol Composition Analysis

#### 2.4.1. Sample Preparation

Two different extraction procedures were employed.

For a hydrophilic extraction, a portion of the PEMC sample (80 mg) was treated with 1 mL of ethanol. The Eppendorf tube was vortexed and incubated in a sonication bath at 45 °C for 60 min. The liquid phase was filtered and lyophilized.

For a lipophilic extraction, the portion of 80 mg was treated with 1 mL of acetonitrile and 500 μL of benzene. The Eppendorf tube was vortexed and incubated in a sonication bath at 45 °C for 60–105 min. The liquid phase was filtered and lyophilized.

10 mg of dry products from both hydrophilic and lipophilic extractions were dissolved in the LCMS-grade acetonitrile to produce the sample concentration of 0.5 mg/mL used for UHPLC/MS analyses.

#### 2.4.2. Qualitative Analysis

Qualitative analysis was performed using a Shimadzu Ultra-High-Performance Liquid Chromatograph (Nexera XR LC 40) coupled to an MS/MS detector (LCMS 8060, Shimadzu Italy, Milan, Italy). The MS/MS was operated with electrospray ionization (ESI) and controlled by Lab Solution software, which simultaneously provided quick switching from a low-energy scan at 4 V (full scan MS) to a high-energy scan (10–60 V ramping) during a single LC run. The source parameters were set as follows: nebulizing gas flow 2.9 L/min, heating gas flow 10 L/min, interface temperature 300 °C, DL temperature 250 °C, heat block temperature 400 °C, and drying gas flow 10 L/min. The analysis was performed by flow injection with the mobile phase consisting of acetonitrile: water + 0.01% formic acid (5:95, *v*/*v*). The instrument was set for a SIM experiment in negative mode (only syringic acid in positive ESI). Sample peaks were considered “positive” if the area under them was higher in magnitude than that of the blank (See Appendix A).

### 2.5. Animals

In this study, we used both sexes of Wistar rats weighing between 150–200 g and Swiss albino mice weighing between 23–29 g. The animals were acquired from the animal facility at the Department of Biology, at the Faculty of Sciences Dhar el Mahraz, Fez, Morocco.

Before each test, the selected animals were subjected to an acclimatization period of 14 days with controlled parameters of temperature (22 °C ± 2 °C), humidity (45–50%), and 12:12 h light–dark cycle. The study was conducted according to the guidelines of the Declaration of Helsinki, and approved by the Institutional Review Board at the Faculty of Sciences, Dhar el Mahraz, Fez, Morocco (02/17-LBEAS-04 and 03/01/2020).

### 2.6. Antileukemic Activity

#### 2.6.1. Cell Culture

Human cancer and Vero cell lines were obtained from the National Institute of Amazonian Research, Manaus (AM), Brazil. HL60 (ATCC^®^ CCL-240^TM^ Human acute promyelocytic leukemia), K562 (ATCC^®^ CCL-243^TM^—human chronic myelogenous leukemia) and Vero cell lines (2 × 10^4^ cells per well) were cultured into a 96-well plate containing 0.2 mL per well of the RPMI medium supplemented with 10% FBS, penicillin-streptomycin, and fungizone, in an atmosphere of 5% CO_2_ at 37 °C for 24 h. After the formation of a sub-confluent monolayer, the cells were treated with different concentrations of the PEMC and incubated again at the same conditions for 24, 48, and 72 h.

#### 2.6.2. Cytotoxicity Assay

We assessed the cytotoxicity of the PEMC by the MTT assay. The HL60, K562, and Vero cell lines (2 × 10^4^ per well) were cultured into a 96-well plate containing 0.2 mL per well of the RPMI medium supplemented with 10% FBS, penicillin-streptomycin, and fungizone, in an atmosphere of 5% CO_2_ at 37 °C for 24 h.

Once the sub-confluent monolayer had been formed, the cells were treated with different concentrations of the PEMC (diluted in PBS with 0.5% of DMSO) and incubated again at the same conditions for 24, 48, and 72 h. Sterile PBS and DMSO 0.5% were used as a negative control and DMSO 100% as a positive one. Subsequently, the medium was removed from all wells, and 10 µL of the MTT (5 mg/mL in sterile PBS) diluted in 100 µL of the DMEM medium (without a phenol red to avoid misinterpretation) was added into the wells and incubated for 4 h at the same conditions as stated above. After that, the MTT was removed, and 50 µL of MTT lysis buffer was added to each well. The content was gently homogenized to dissolve formazan crystals and incubated for 10 min at 37 °C. All assays were performed in triplicate. Optical densities of the samples were measured on a microplate reader at a wavelength of 570 nm. The relative viability of the cells was estimated using the following equation:$$Cell\;viability=\frac{A570\;\mathrm{of}\;\mathrm{the}\;\mathrm{treated}\;\mathrm{sample}}{A570\;\mathrm{of}\;\mathrm{the}\;\mathrm{untreated}\;\mathrm{sample}}\times 100$$

### 2.7. In Vitro Hemolysis, Hemolytic Effect of the PEMC

The hemolytic effect of the polyphenol-enriched fraction under investigation was evaluated according to a slightly modified method of Li and Liu [45]. Blood samples from rats were collected in heparinized tubes. After centrifugation at 1500 rpm for 5 min, the supernatant was removed, the pellet was washed three times with the PBS (125 mM NaCl, 10 mM sodium diphosphate, pH 7.4) and centrifugated. The last centrifugation lasted 10 min.

The cell pellet was diluted with the PBS solution to obtain a hematocrit of 2%. The PEMC was diluted in PBS to obtain different concentrations of 20, 10, 5, and 1 mg/mL. In each tube, 1 mL of the PEMC at different concentrations, 2.5 mL of the PBS, and 3.5 mL of the prepared erythrocyte suspension were added. The tubes were mixed gently and left in a shaker incubator at 37 °C for 60 min.

After centrifugation at 1500 rpm for 5 min, the absorbance of each tube was read at 630 nm with a UV-Vis spectrophotometer against a blank containing PBS. A negative control tube was prepared in the same experimental steps, composed of erythrocyte suspension and the PBS buffer solution in the absence of the PEMC. The experiment was repeated three times, and the rate of hemolysis at different PEMC concentrations was calculated as a percentage of total hemolysis according to the following formula:$$\mathrm{Hemolysis}\;\mathrm{rate}\;\left(\%\right)=\frac{Abs_{extract}-Abs_{negative\;control}\;}{Abs_{total\;hemolysis}}\times 100\%$$

### 2.8. Anti-Inflammatory Activity

#### 2.8.1. In Vivo Carrageenan-Induced Edema Test

This test was performed following the method of Winter and Porter [46]. Injection of carrageenan under the plantar fascia of the hind paw of a rat resulted in an edema in the metatarsal region. Intensity of the edema was assessed by the increase in the volume of the paw as a percentage of the initial volume. Preventive administration of an anti-inflammatory product significantly reduces the edema development.

The rats were divided into four groups (5 rats in each group) homogeneous in weight:

Group 1: animals receiving the PEMC at 50 mg/kg b.w.

Group 2: animals receiving the PEMC at 100 mg/kg b.w.

Group 3: animals receiving placebo only (distilled water).

Group 4: reference group receiving diclofenac at 10 mg/kg b.w.

1% solution of carrageenan in distilled water was used in the experiment. The circumference of the left hind leg was measured for each rat before the carrageenan injection. The doses of the PEMC treatment, diclofenac, and placebo were administered by oral gavage. One hour after the gavage, each rat received an injection of 50 µL of 1% carrageenan solution under the plantar pad of the left hind paw to induce edema. The edema evolution was monitored by measurement of its volume 1, 3, 4, 5, and 6 h after the carrageenan injection. The edema inhibition percentage was calculated according to the formula:$$\mathrm{nhibition}\;\left(\%\right)=\frac{Average\;{\mathrm{volume}}_{control\;group}-Average\;{\mathrm{volume}}_{test\;group}}{Average\;{\mathrm{volume}}_{control\;group}}\times 100\%$$

#### 2.8.2. In Vivo Wound Healing Activity

##### A. Formulation of Ointments

The ointments containing the polyphenol-enriched fraction extracted from leaves of the *M. communis* L. were prepared by mixing together accurately weighed amounts of the PEMC and a petrolatum (vaseline) used as an excipient. The latter was added in several steps by portions. The mixture was thoroughly homogenized after the addition of each portion of the excipient [47].

Two ointments were prepared containing 0.05% and 0.1% of the PEMC, respectively. For the PEMC-0.05% ointment, 25 mg of the PEMC were mixed with 49.975 g of vaseline. For the PEMC-0.1% ointment, 50 mg of the PEMC were mixed with 49.95 g of vaseline.

##### B. Wound Healing Assessment

The animals were divided into 4 groups of 5 rats each, and all treatments were applied topically. A control group received a placebo (Vaseline only) treatment, the other groups were treated with madecassol (1%), PEMC-0.05%, and PEMC-0.1% ointments, respectively.

On the first day (D 0) of the experiment, the animals were anesthetized and the down on the animal’s back was shaved with a sterilized clipper. Subsequently, two circular wounds of 10 mm diameter were created on each side of the spine. During the entire experiment (Day 0 to Day 18), the wounds were cleaned and measured daily.

To evaluate the wound healing potential of the PEMC, the evolution of the wound surfaces was observed every day. The wound size reduction was calculated as follows:$$Reduction\;\left(\%\right)=\frac{Average\;wound\;area_{initial}-Average\;wound\;area_{test}}{Average\;wound\;area_{control}}\times 100\%$$

### 2.9. Antioxidant Activity

#### 2.9.1. β-Carotene Bleaching Test

The antioxidant activity of the PEMC was evaluated by a coupled oxidation of β-carotene and linoleic acid in an aqueous emulsion as described in [48]. The antioxidant capacity was determined by measuring the inhibition of β-carotene oxidative degradation (decolorization) by linoleic acid oxidation products [49].

Kinetics of the emulsion decolorization was monitored at 490 nm at regular time intervals for 48 h both in the presence and in the absence of the antioxidant. Butylated hydroxytoluene (BHT) was used as a reference sample. The test tubes were prepared in parallel with the negative control where the sample was replaced by 350 µL of methanol.

The relative antioxidant activity of the extracts (RAA) was calculated according to the following equation:$$\mathrm{RAA}\;\left(\%\right)=\frac{\mathrm{Absorbance}\;48\;h_{sample}}{\mathrm{Absorbance}\;48\;h_{\mathrm{BHT}}}\times 100\%$$

#### 2.9.2. Assessment of the Ferric Reducing Antioxidant Power (FRAP)

The reducing power of the extracts was determined according to the method of Hong et al. [50]. To a test tube containing 0.1 mL of sample, 2 mL of phosphate buffer (0.2 M, pH 6.6) solution was added followed by 2 mL of potassium hexacyanoferrate [K_3_Fe(CN)_6_] (10 g/L). The preparation was heated in a water bath at 50 °C for 20 min. After that, 2 mL of trichloroacetic acid (100 g/L) was added and the mixture was centrifuged at 3000 rpm for 10 min. Finally, 2 mL of the supernatant was mixed with 2 mL of distilled water and 0.4 mL of ferric chloride FeCl_3_ (1 g/L). A blank without a sample was prepared under the same conditions. The readings were measured at 700 nm. Ascorbic acid was used as a positive control

#### 2.9.3. Total Antioxidant Capacity (TAC)

A volume of 0.3 mL of the PEMC was mixed with 3 mL of a reagent solution (0.6 M sulfuric acid, 28 mM sodium phosphate, and 4 mM ammonium molybdate). The tube was screwed down and incubated at 95 °C for 90 min. After cooling, the absorbance of the solution was measured at 695 nm against the blank containing 3 mL of the reagent solution and 0.3 mL of methanol and incubated under the same conditions. The total antioxidant capacity was expressed in milligram equivalents of ascorbic acid per gram of dry matter (mg-eqv AA/g) [51].

### 2.10. Acute Toxicity Study

The acute toxicity of the polyphenol-enriched fraction extracted from leaves of *Myrtus communis* L. was evaluated according to the OECD Guideline 423 in groups of 5 Swiss Albino mice. The PEMC was administered by gavage in a single oral dose of 50, 100, and 2000 mg/kg body weight. The animals were observed for 14 days after the treatment, they had free access to food and water during the experiment.

After 14 days, the animals were weighed and blood samples were collected for biochemical analysis. Subsequently, the animals were euthanized, and their liver, spleen, and kidneys were collected.

### 2.11. Statistical Analysis

Statistical analyses were performed using a Graph Pad Prism software version 8.0. Data were expressed as means ± standard deviation. The results were analyzed using analysis of variance and Tukey as a post hoc test. The differences of cell viability observed for treatments with different PEMC concentrations were evaluated using *t*-test and ANOVA. The IC_50_ was estimated using non-linear regression. The results were considered statistically significant at *p* < 0.05.

## 3. Results

### 3.1. Polyphenol Analysis

The UHPLC/MS-MS analysis revealed the presence of seventeen components in the prepared polyphenol-enriched fraction, namely: gallic acid, quercetin, *p*-coumaric acid, hesperidin, amentoflavone, luteolin, quercetin-3-O-glucoside, quercetin-3-O-glucuronic acid, isorhamnetin-7-O-pentose, luteolin 7-O-glucoside, kaempferol-3-O-glucuronic acid, kaempferol-3-O-pentose, kaempferol-3-O-hexose deoxyhexose, catechin gallate, procyanidin, kaempferol, and naringin (Table 1). Relative abundance of the constituents in the fraction was estimated on the peak area basis by comparison of those of the sample with the blank (see Appendix A). Quercetin-3-O-glucoside, isorhamnetin-7-O-pentose, and luteolin 7-O-glucoside were the most abundant molecules in the PEMC. It was also noted that most of the constituents (viz., 15 of 17) belong to flavonoids, therefore, the fraction may be considered as a flavonoid-rich one.

### 3.2. Antileukemic Activity

The cytotoxic activity of the PEMC was evaluated using three cell lines: human acute promyelocytic leukemia (HL60 cell line), human chronic myelogenous leukemia (K562 cell line), and normal Vero cells. The cytotoxicity indices were estimated as cell viability percentage measured by the MTT assay in a dose-dependent manner after 24, 48, and 72 h of treatment with increasing doses (0 to 100 µg/mL) of the PEMC.

The polyphenol-enriched fraction was able to inhibit the proliferation of HL60 (IC_50_ = 19.87 µM) and K562 (IC_50_ = 29.64 µM) cancerous cell lines. The 50% reduction in the cell viability was found after 48 h at the dose of 100 µg/mL for both cancerous cells (Figure 1). For the HL60, the maximum cytotoxicity of 80% was reached at 50 µg/mL concentration after 72 h, while for the K562 cells the maximum cytotoxicity of 60% was noted for 100 µg/mL concentration after 48 h treatment. No cytotoxicity effect was observed for Vero cells at all concentrations tested (IC_50_ > 100 µM) (Figure 2).

### 3.3. In Vitro Hemolysis

To study the hemolytic effect of the polyphenol-enriched fraction extracted from *Myrtus communis* L., the effect was tested on erythrocytes isolated from blood samples collected from rats. The results shown in Figure 3 illustrate the evolution of the hemolysis rate of the red blood cells as a function of the PEMC concentration. The hemolysis ratio less than 5% indicates that the fraction was non-hemolytic, and therefore may be safe for use.

### 3.4. Anti-Inflammatory Activity

#### 3.4.1. In Vivo Carrageenan-Induced Edema Test

Figure 4 shows the PEMC effect on carrageenan-induced inflammatory edema on the left paw of the rats. Oral administration of diclofenac (10 mg/kg) prevented significantly the increase in the left paw volume with 42.61, 57.22, 68.75, 80.94, and 94.7 inhibition percentages after 1, 3, 4, 5, and 6 h of the carrageenan injection, respectively. Oral administration of the PEMC in doses of 50 and 100 mg/kg b.w. resulted in a decrease in the edema at a rate of 23.49, 44.44, 62.5, 78.57, and 89.41% (for 50 mg/kg b.w.) and 48.08, 68.25, 81.25, 95.23 and 100% (for 100 mg/kg b.w.) after 1, 3, 4, 5, and 6 h, respectively. The data obtained confirm that PEMC at the dose of 100 mg/kg b.w. exhibits a remarkable anti-inflammatory activity exceeding that of diclofenac. The anti-inflammatory activity maximum was observed 5 h after the edema induction.

#### 3.4.2. In Vivo Wound Healing Activity

According to the results obtained, the ointments containing 0.05% and 0.1% of PEMC reduced significantly the wound surface area (Figure 5) as a function of time, when compared to the control treatment. The wound healing acceleration was observed, with the maximum (≈100%) reached on the twelfth day of the treatment. Remarkably, the positive control group receiving the madecassol (1%) ointment arrived to the maximum value (≈100%) only on the 18th day, while the placebo group reached the maximum of the wound healing process only after the 18th day.

### 3.5. Antioxidant Activity

#### 3.5.1. β-Carotene Bleaching Test

The PEMC inhibit the coupled oxidation of linoleic acid and β-carotene compared to the negative control as shown in Figure 6. However, it was less effective than the BHT (butylated hydroxytoluene).

A bleaching delay observed in the test indicates that the PEMC may perform as a free radical scavenger [16] protecting the β-carotene against free radicals generated by linoleic acid peroxidation. The antioxidant effect could be explained by the presence of polyphenols in the fraction.

#### 3.5.2. Ferric Reducing Antioxidant Power (FRAP)

The antioxidant activity of polyphenolic extracts of *Myrtus Communis* L. (PEMC) was also evaluated using the FRAP method. It is a universal, simple, rapid, and reproducible assay that can be applied either in organic or aqueous extracts [52]. The presence of reductants in plant extracts causes the reduction in Fe^3+^ in a ferricyanide complex to the ferrous form. Therefore, Fe^2+^ can be assessed by measuring and monitoring the increase in blue color density in the reaction medium at 700 nm.

The results obtained show that the reducing power of the PEMC is dose-dependent and it is much lower (EC_50_: 3.033 ± 0.378 EAA/g) than that of quercetin (EC_50_: 0.032 ± 0.002 EAA/g).

#### 3.5.3. Total Antioxidant Capacity (TAC)

The total antioxidant capacity of the PEMC is 0.171 ± 0.003 mg EAA/g (mg of ascorbic acid equivalent per g of the dry fraction) indicate a good antioxidant activity.

### 3.6. In Vivo Acute Toxicity Study

#### 3.6.1. Body Weight and General Aspect

A single oral administration of the PEMC at the dose of 50 or 100 mg/kg did not disturb the mice growth (Figure 7). It is interesting to note that even a single oral administration of the PEMC at the dose of 2000 mg/kg did not disturb the growth of the animals and showed no toxic effect. The results suggest that the PEMC is non-toxic, since no mortality or changes in general condition were observed in mice that received the dose of 2000 mg/kg.

#### 3.6.2. Biochemical Analysis

The results of biochemical analysis of blood serum taken from the mice groups treated orally with the doses of 50, 100, and 2000 mg/kg showed no significant change in the levels of ASAT and ALAT transaminases suggesting that the PEMC is not hepatotoxic. Additionally, the urea and creatinine values relating to the health of kidneys did not altered within the study, even for the highest dose of 2000 mg/kg (Table 2).

#### 3.6.3. Effect of Acute Administration of the PEMC on the Relative Weight of Organs

The acute oral toxicity of the PEMC effect on different vital organ weights was evaluated in mice. At the end of the experiment, kidneys, livers, and spleens were collected from different mice groups and weighed. The average weights of the internal organs in question had no significant deviations when compared to the normal control (Table 3). Those data are in good agreement with the biochemical analysis described above. Moreover, no gastric toxicity was found by a macroscopic stomach observation.

## 4. Discussion

Our findings demonstrated a significant specific cytotoxicity effect of the PEMC towards the cancerous cells (HL60 and K652) without affecting the non-cancerous ones (Vero). When studying anticancer properties of natural compounds, it is very important to consider antioxidant activity as well. Oxidative stress is responsible for causing many different diseases, including cancer, especially because of an imbalance between the formation and neutralization of prooxidant moieties. Free radicals initiate the oxidative stress that ultimately cause protein and DNA damage and may provoke a peroxidation of lipids as well [1,2]. Those physiological changes contribute to carcinogenesis. The antioxidant activity of phenolic compounds is related to their redox characteristics, which enable them to function as reducing agents, hydrogen donors, free radical scavengers, singlet oxygen quenchers, and metal chelating agents, among other functions [53]. It was shown that a total phenolic content in a variety of seeds, fruits, and vegetables directly correlates with the antioxidant activity [54]. The present study demonstrated the antioxidant role of the PEMC through three different methods (TAC, FRAP and β-carotene bleaching), confirming its action as either a free radical scavenger or reducing agent. Those findings support the pharmaceutical activity of the PEMC as a prophylactic anticancer agent.

In this study, we identified seventeen compounds in the PEMC of *Myrtuus communis,* fifteen of them were flavonoids. Quercetin is found in a variety of edible plants. It possesses antihypertensive, anticarcinogenic, anti-inflammatory, antiulcer, and antiviral properties [55]. It is one of the most studied flavonoids in epigenetic research, with numerous studies attempting to uncover its anticancer potential on various cancer cell lines, such as MCF-7 and MDA-MB-231 [56], colon cancer [57], osteosarcoma [58] and many other. Tseng et al. investigated DNA fragmentation, PARP, and procaspases activities to discover that quercetin might induce apoptosis in HL-60 human leukemia cells in a dose-dependent manner [59]. Luteolin is a flavone found in many fruits and vegetables. It demonstrated a capability to decrease the viability of lung (LNM35), colon (HT29), liver (HepG2), and breast (MCF7/6 and MDA-MB231-1833) cancer cells mainly through an inhibition of histone deacetylases (HDAC) [60]. Gallic acid is an phenolic acid widely distributed in plants. It has protective effects against human cancers by targeting DNA methyltransferases (DNMTs) [61,62]. Amentoflavone is a biflavonoid found in many plants, such as *Ginkgo biloba*, *Chamaecyparis obtusa*, *Xerophyta plicata,* etc. [63]. It was reported that amentoflavone can reduce cell viability and induce apoptosis in glioma cell lines in a dose-dependent manner [64]. Isorhamnetin-7-O-pentose, quercetin-3-O-glucoside, and luteolin 7-O-glucoside were the most abundant components of the PEMC fraction. Although only a few studies of the pharmacological potential of the isorhamnetin-7-O-pentose have been reported, the flavonoid shows a great therapeutic potential for type 2 diabetes by down-regulating the protein tyrosine phosphatase-1B (PTP1B) expression [65]. The quercetin-3-O-glucoside demonstrated anti-inflammatory and antileukemic activities by inhibiting cyclooxygenase-1/2 pathway and inducing apoptosis, respectively [66,67]. The luteolin 7-O-glucoside is a glycosyloxyflavone that acts as an antioxidant and an anti-inflammatory agent through a regulation of inflammatory mediators and oxidative stress [68,69,70]. Additionally, the luteolin 7-O-glucoside may induce apoptosis of nasopharyngeal carcinoma cells via the AKT signaling pathway [71].

Anticancer drugs that demonstrate anti-inflammatory activities tend to be more reliable for tumor treatment, since inflammation plays an important role in the formation of carcinogenic state [3]. We assessed the anti-inflammatory activity of the PEMC using both in vitro (hemolysis) and in vivo (carrageenan-induced edema and wound healing) assays. The carrageenan-induced paw edema is a very sensitive and reproducible test for nonsteroidal anti-inflammatory medications, which has been used for a long time to look for novel anti-inflammatory treatments [72]. The assay helps to identify active compounds through oral administration and has a high predictive value for anti-inflammatory medicines operating via acute inflammation mediators [73], since the carrageenan injection results in an immediate and localized inflammatory response. Histamine, serotonin, and bradykinin are the first mediators to be implicated in the early phase (0–1 h), while prostaglandins and other cytokines such as IL-1, IL-6, IL-10, and TNF are implicated in the second phase [72,73]. Increased vascular permeability, increased blood flow, and infiltration of neutrophils and macrophages are hallmarks of an acute inflammatory response. Furthermore, exudation of fluid and plasma proteins together with a buildup of leukocytes near the inflammatory site often result in edema, which was treated efficiently by the polyphenol-enriched fraction. It is interesting to note that the PEMC at the dose of 100 mg/kg demonstrated the best anti-inflammatory effect throughout the study. It was even superior than the diclofenac used as the positive control.

As the PEMC showed a pronounced anti-inflammatory activity, we checked its wound healing potential, bearing in mind that those two biological activities are intrinsically correlated. Wound healing is a dynamic process involving numerous biochemical reactions that aim to restore an injured cellular structure to its original condition. Inflammation, proliferation, and remodeling are three consecutive and overlapping stages in the traditional wound healing cascade [74]. Topical administration of the prepared ointments (0.1 and 0.05%) of the PEMC demonstrated a powerful wound healing effect in compassion with both positive (madecassol) and negative controls.

Hemolysis studies demonstrated stability of red blood cells when contacting with a foreign substance (i.e., the PEMC), and thus the fraction was considered non-hemolytic (hemolysis rate was below 5%) according to the biological safety guidelines [75].

Safety and tolerability of chemotherapy have a significant influence on patients’ quality of life and may sometimes dissuade them from continuing the treatment [76]. Prescribed tyrosine kinase inhibitors, such as imatinib and others, have been linked to liver damage [77], chronic tiredness [78], nausea, rash, superficial edema, muscle cramps, and myelosuppression [79] similar to chronic myeloid leukemia therapy. The findings of both in vivo and in vitro experiments show that the PEMC has no toxicity, which enables to focus more attention on its antileukemic activity.

## 5. Conclusions

Our findings demonstrated that the PEMC has a selective cytotoxicity effect against leukemia cell lines, since no toxicity towards the non-cancerous Vero cell line was observed. Furthermore, the toxicological experiments in vivo and in vitro clearly demonstrated the absence of toxicity, which makes the polyphenol-enriched fraction a good candidate for prospective studies on pharmaceutical approaches. Besides the antileukemic activity, this study also revealed a high anti-inflammatory, antioxidant, and wound healing potential of the PEMC; thus, the polyphenol-enriched fraction could be a new effective and safe alternative to current treatments. However, additional research related to the profile of chemokines, growth factors, and cytokines elicited by the PEMC is necessary to better characterize the rich pharmaceutical potential of the polyphenol-enriched fraction extracted from leaves of *M. communis*.

## Figures and Tables

**Figure 1 nutrients-14-05055-f001:**
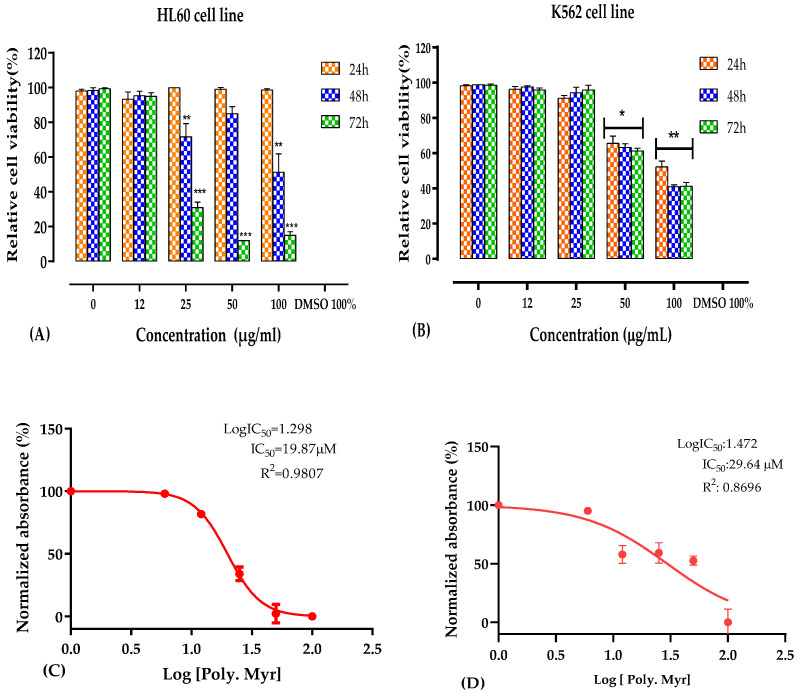
Cytotoxicity of the polyphenol-enriched fraction extracted from *Myrtus communis* for K562 and HL60 cells. (**A**) K562 and (**B**) HL60 cell viability after 24–72 h of treatment with different concentrations of the PEMC (12–100 µg/mL). The IC_50_ values for K562 (**C**) and HL-60 (**D**) were estimated using nonlinear regression. The absorbance values were measured at the wavelength of 570 nm, and the mean values ± SD of three experiments are displayed along with a representative IC_50_ curve. The cell viability was estimated by the MTT assay. * *p* < 0.05; ** *p* < 0.01; *** *p* < 0.001.

**Figure 2 nutrients-14-05055-f002:**
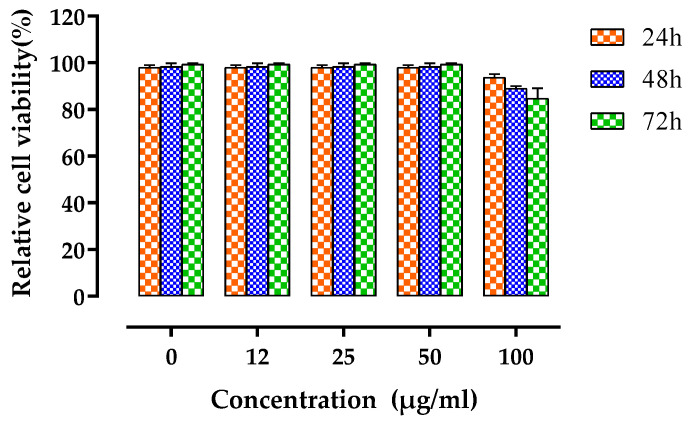
Cytotoxicity of the polyphenol-enriched fraction extracted from *Myrtus communis* for the normal Vero cell line. Relative viability of Vero cells after 24–72 h of treatment with different concentrations of the PEMC (12–100 µg/mL). The absorbance values were measured at the wavelength of 570 nm, and the mean values ± SD of three experiments, the cell viability was estimated by the MTT assay.

**Figure 3 nutrients-14-05055-f003:**
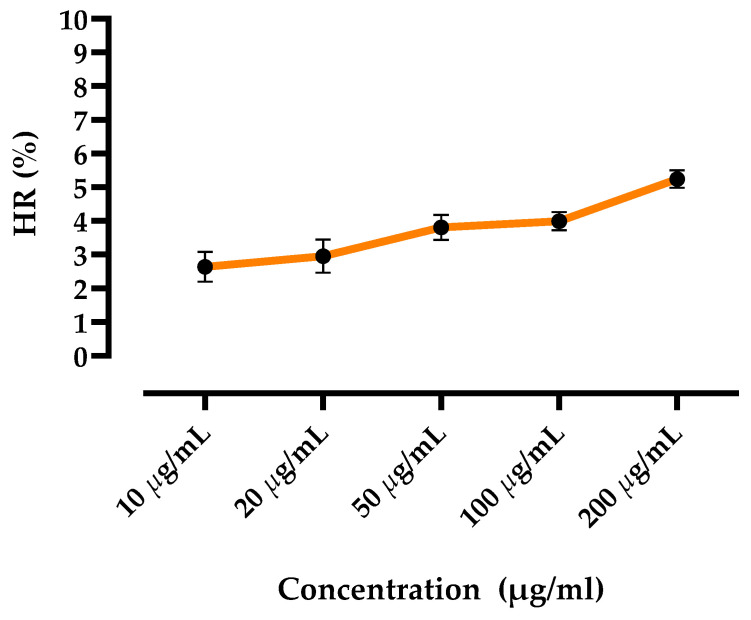
Evolution of the hemolysis level. The results are expressed as mean ± standard deviation. The assays were performed in triplicate. HR = Hemolysis ratio.

**Figure 4 nutrients-14-05055-f004:**
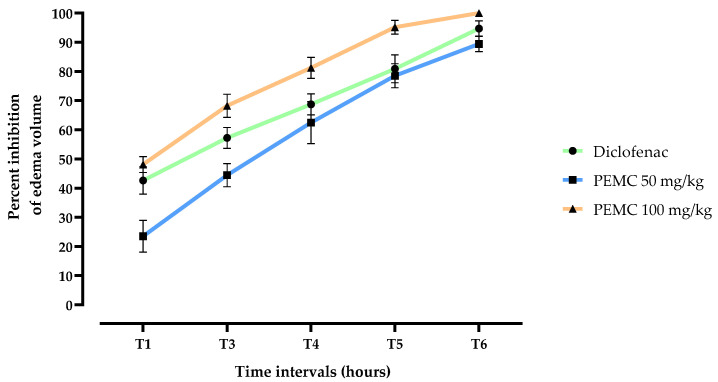
Inhibition percent of the edema volume after the treatment with PEMC. Results are expressed as mean ± standard deviation. The experiments were performed in quintuplicate.

**Figure 5 nutrients-14-05055-f005:**
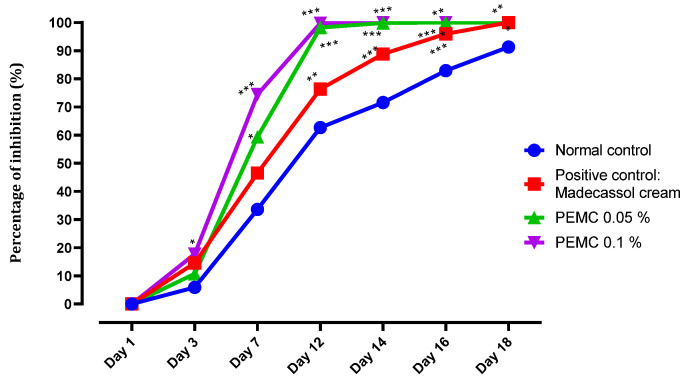
Wound healing assessment after treatment with PEMC containing ointments. The results are expressed as mean ± standard deviation. The tests were done in quintuplicates; * *p* < 0.05; ** *p* < 0.01; *** *p* < 0.001 compared to the normal control.

**Figure 6 nutrients-14-05055-f006:**
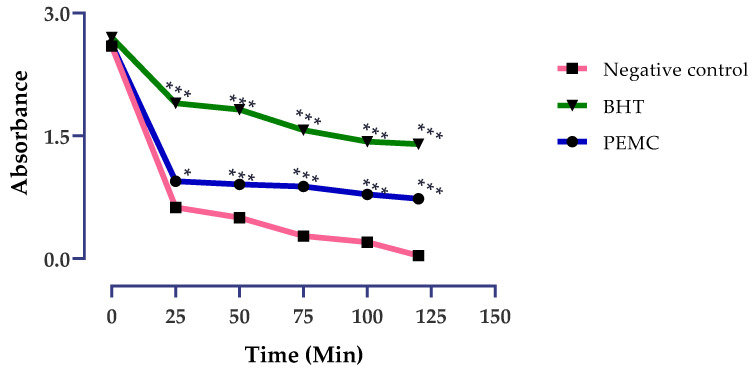
Antioxidant activity of the PEMC, the results are expressed as mean ± standard deviation. The tests were realized in triplicate. * *p* < 0.05; *** *p* < 0.001 compared to the normal control.

**Figure 7 nutrients-14-05055-f007:**
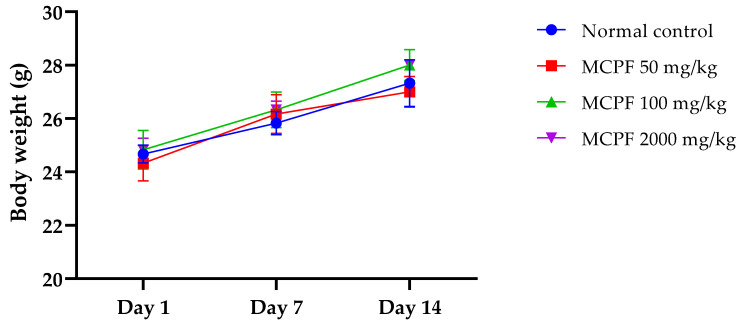
Bodyweight development of mice treated with a single oral administration of the PEMC.

**Table 1 nutrients-14-05055-t001:** Constituents identified in the polyphenol-enriched fraction extracted from *Myrtus communis* and their relative abundances.

Constituent	[M − H]^−^	Relative Abundance
gallic acid	168.9	++
quercetin	301	+
*p*-coumaric acid	162.9	+
hesperidin	301.3	+
amentoflavone	537.1	++
luteolin	284.9	+
quercetin-3-O-glucoside	463.1	+++
quercetin-3-O-glucuronic acid	477	+
isorhamnetin-7-O-Pentose	447.1	+++
luteolin 7-O-glucoside	447	+++
kaempferol-3-O-glucuronic acid	461.1	+
kaempferol-3-O-pentose	417.1	+
kaempferol-3-O-hexose deoxyhexose	593.1	+
catechin gallate	441	+
procyanidin	577	+
kaempferol	285	+
naringin	579	+

Relative peak intensities: + medium, ++ strong, +++ very strong.

**Table 2 nutrients-14-05055-t002:** Biochemical analysis on hepatic and renal biomarkers in the serum of mice treated with different doses of the polyphenol-enriched fraction extracted from *Myrtus communis*.

	Normal Control	PEMC 50 mg/kg	PEMC 100 mg/kg	PEMC 2000 mg/kg
ALAT	36.80 ± 1.11	37.67 ± 1.76	41.33 ± 2.6	33.67 ± 2.48 *
ASAT	307.7 ± 20.37	284.33 ± 17.12	318.67 ± 26.76	304 ± 31.33 **
Urea	0.28 ± 0.02	0.26 ± 0.02	0.3 ± 0.01	0.28 ± 0.02 *
Creatinine	3.40 ± 0.31	3.93 ± 0.6	3.3 ± 0.24	3.6 ± 0.34 *

The results are expressed as mean ± standard deviation. The assays were performed in triplicates; * *p* < 0.05, ** *p* < 0.01 compared to normal control.

**Table 3 nutrients-14-05055-t003:** The PEMC effect on relative weight (g) of several vital organs.

	Normal Control	PEMC 50 mg/kg	PEMC 100 mg/kg	PEMC 2000 mg/kg
Liver (g)	7.15 ± 0.73	6.34 ± 0.41	6.41 ± 0.44	6.92 ± 0.51
Kidneys (g)	1.19 ± 0.24	1.12 ± 0.12	1.17 ± 0.14	1.21 ± 0.13
Spleen (g)	0.44 ± 0.63	0.46 ± 0.04	0.47 ± 0.05	0.42 ± 0.09

Results are expressed as mean ± standard deviation. Experiments were performed in triplicates.

## Data Availability

Data are available upon request.

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
