# Peer review of "Antileukemic, Antioxidant, Anti-Inflammatory and Healing Activities Induced by a Polyphenol-Enriched Fraction Extracted from Leaves of Myrtus communis L."

_nutrients, 2022, doi:10.3390/nu14235055_

Round 1

Reviewer 1 Report

The study is more or less very formal. Eventual hemolytic effect do not fit to anti-inflammatory activity. The mentioned activity does not correspond obviously to antileukemic effectiveness.

In addition, experimental data presented in Fig. 1 are badly fitting to the "concentration-response" curve in HL-60 leukemic cells, which are expected to be more sensitive than K-562 cells.

There are no reference experimental data about comparison with pure active compounds.

The phytochemical analysis of the extract is not complete and presented data are insufficient. 

The manuscript does not contain any substantial data about the eventual mechanism of action, e.g. induction of apoptosis, caspase activation etc.

Author Response

We thank the Reviewer for the comments. Please, find the answers to the Reviewer's comments below.

  1. The study is more or less very formal. Eventual hemolytic effect do not fit to anti-inflammatory activity. The mentioned activity does not correspond obviously to antileukemic effectiveness.

Response: We agree that the hemolytic effect was misplaced in the anti-inflammatory section. The mistake was corrected. We also agree that the results do not confirm the antileukemic effectiveness of the PEMC in detail, since we have not checked it. To assess the antileukemic effectiveness of polyphenol-enriched fraction, more specific and detailed experiments using appropriate in vivo and in vitro models would be required. However, those were not the objectives of the study. The objective was to show the antileukemic, anti-inflammatory, and healing potential of the polyphenol-enriched fraction extracted from leaves of M.communis. Our results show that the PEMC demonstrated a great potential as a promising prophylactic and therapeutic agent, especially against leukemia and skin lesions, to be explored in further pharmaceutical studies. We also changed the title of the manuscript to better represent our investigation.

  1. In addition, experimental data presented in Fig. 1 are badly fitting to the "concentration-response" curve in HL-60 leukemic cells, which are expected to be more sensitive than K-562 cells.

Response: Different susceptibility of cancerous cells to substances or extracts depend on their anticancer mechanism of action. Our results demonstrated that HL60 cells were more sensitive to the PEMC, than the K562 ones. As shown in the figure 1C, the anticancer activity started to be observed at the concentration of 25 µg/ml in HL60 cells, while for the K562 cells, the cytotoxicity activity begun at the concentration of 50 µg/mL. The above is clearly confirmed by statistical analysis performed through Dunnet’s test and non-linear regression, as demonstrated in the graph (HL60: LogIC50=1.298, IC50=19.87µM; K562: LogIC50=1.33, IC50=21.24µM and R2= 8696).

  1. There are no reference experimental data about comparison with pure active compounds.

Response: We thank the reviewer for the comment. The discussion is provided (please, see p.14).

  1. The phytochemical analysis of the extract is not complete and presented data are insuficiente

Response: Phytochemical analysis of the Myrtus communis was not the objective of the study, taking into a consideration that the plant is well-known. Our objective was to study antileukemic, antioxidant, anti-inflammatory, and healing activities induced by the polyphenol-enriched fraction extracted from leaves of the plant. Analysis of that fraction is complete and supported also by the Supplementary material.

  1. The manuscript does not contain any substantial data about the eventual mechanism of action, e.g. induction of apoptosis, caspase activation etc.

Response: We agree that our results just show the potential antileukemic activity of the PEMC, but do not demonstrate the mechanism of its action. Firstly, because elucidation of the mechanism in question was not the goal of the present investigation. Secondly, we did not have enough budget. The latter constraints did not allow us to unveil the mechanism involved in the anticancer activity observed. However, we have an intention to clarify that mechanism in future studies.

Reviewer 2 Report

The title must be changed, clarifiying both in title as well in the text that htese are in vitro experiments and in vivo NOT in human beings. As such, better fly low...if it will be done, I have no major concerns on this paper.

Author Response

We thank the Reviewer for the comments. Please, find the answers to the Reviewer's comments below.

Comment: The title must be changed, clarifiying both in title as well in the text that htese are in vitro experiments and in vivo NOT in human beings. As such, better fly low...if it will be done, I have no major concerns on this paper.

Response: The title of the manuscript was changed to better reflect the study, as suggested.

Reviewer 3 Report

Reviewers' Comments to Authors:

A manuscript entitled "
Prophylactic and Therapeutic Antileukemic Activities Induced by a Polyphenolic Extract From Myrtus Communis L” describes the medicinal properties of the extract of Myrtus Communis L and causes a great interest. Without doubt, this manuscript has merit and may be accepted, however, there are some omissions and significant drawbacks with regards to the submitted manuscript. Further, the professional proofreading will substantially improve the quality of this paper as there are glaring grammatical errors throughout. In addition, expression “Polyphenolic extract” is not correct in many reasons. For example, the described extract may contain other compounds that are not polyphenols.    

 The introduction part doesn’t contain any information about wound healing problems, in this case such publications may be recommended:

1.      Budovsky A., Yarmolinsky L., Ben-Shabat S. (2015). Effect of medicinal plants on wound healing. Wound Repair and Regeneration, 23, 171-183.

2.       Han G, Ceilley R. (2017). Chronic Wound Healing: A Review of Current Management and Treatments. Adv Ther;34(3):599-610. doi: 10.1007/s12325-017-0478-y.

3.       Rezaie F, Momeni-Moghaddam M, Naderi-Meshkin H. (2019). Regeneration and Repair of Skin Wounds: Various Strategies for Treatment. Int J Low Extrem Wounds. 18(3):247-261.

As already mentioned, this manuscript has several drawbacks.

1.The title does not reflect the performed research perfectly.
2. As already mentioned, the introduction should be improved. It must provide comprehensive analysis of the existent medical problems and explain why medicinal plant investigation may be important and effective. In addition, the aims of the study must be written clearly.
3. Formulation of ointments should be described.
4.
Discussion doesn’t have any comparison of the identified compounds (Table 1) with literature data. It is known that many the identified compounds can act on various biochemical pathways hence causing anti-inflammatory, antioxidant, anti-carcinogenic effects. Such changes in the discussion will allow to analyze the possible mechanisms of the extract’s action.

        5. There are many spelling mistakes and grammatical errors.

Author Response

We thank the Reviewer for the comments. Please, find the answers below.

Comment: A manuscript entitled " Prophylactic and Therapeutic Antileukemic Activities Induced by a Polyphenolic Extract From Myrtus Communis L” describes the medicinal properties of the extract of Myrtus Communis L and causes a great interest. Without doubt, this manuscript has merit and may be accepted, however, there are some omissions and significant drawbacks with regards to the submitted manuscript. Further, the professional proofreading will substantially improve the quality of this paper as there are glaring grammatical errors throughout. In addition, expression “Polyphenolic extract” is not correct in many reasons. For example, the described extract may contain other compounds that are not polyphenols.

Response: We agree with the Reviewer that “Polyphenolic extract” is not correct in many reasons. Nevertheless, the term has been widely used in the literature.

In order to improve the text and to use a more correct description, we modified the expression as “polyphenol-enriched fraction”.

Comment: The introduction part doesn’t contain any information about wound healing problems, in this case such publications may be recommended:

  1. The introduction part doesn’t contain any information about wound healing problems, in this case such publications may be recommended:
  2. Budovsky A., Yarmolinsky L., Ben-Shabat S. (2015). Effect of medicinal plants on wound healing. Wound Repair and Regeneration, 23, 171-183.
  3. Han G, Ceilley R. (2017). Chronic Wound Healing: A Review of Current Management and Treatments. Adv Ther;34(3):599-610. doi: 10.1007/s12325-017-0478-y.
  4. Rezaie F, Momeni-Moghaddam M, Naderi-Meshkin H. (2019). Regeneration and Repair of Skin Wounds: Various Strategies for Treatment.Int J Low Extrem Wounds. 18(3):247-261.

  Response: The introduction was improved, as suggested.

As already mentioned, this manuscript has several drawbacks.

1.The title does not reflect the performed research perfectly.

Response: As suggested, we revised the title to better reflect the study.

  1. As already mentioned, the introduction should be improved. It must provide comprehensive analysis of the existent medical problems and explain why medicinal plant investigation may be important and effective. In addition, the aims of the study must be written clearly.

       Response: Introduction was improved and the aim of the study was rewritten following the suggestions of the Reviewer.

  1. Formulation of ointments should be described.

Response: The corresponding item (2.8.2) was revised according to the suggestion.

  1. Discussion doesn’t have any comparison of the identified compounds (Table 1) with literature data. It is known that many the identified compounds can act on various biochemical pathways hence causing anti-inflammatory, antioxidant, anti-carcinogenic effects. Such changes in the discussion will allow to analyze the possible mechanisms of the extract’s action.

Response: The discussion was improved, as recommended.

  1. There are many spelling mistakes and grammatical errors.

Response: The revision of English was made, as suggested.

Reviewer 4 Report

The manuscript entitled "Prophylactic and Therapeutic Antileukemic Activities Induced by a Polyphenolic Extract From Myrtus Communis L." brings many new, important and promising results for therapy and especially in oncology. I would have the following observation: please explain (improve) the preparation of polyphenolic extracts 2.3. and 2.4.1 and how they were used, for which therapeutic activities.

Author Response

We thank the Reviewer for the comment. Please, find the answer below.

Comment: The manuscript entitled "Prophylactic and Therapeutic Antileukemic Activities Induced by a Polyphenolic Extract From Myrtus Communis L." brings many new, important and promising results for therapy and especially in oncology. I would have the following observation: please explain (improve) the preparation of polyphenolic extracts 2.3. and 2.4.1 and how they were used, for which therapeutic activities.

Response: The items 2.3 and 2.4.1 were improved, as recommended. The PEMC was used to assess the antileukemic, antioxidant, anti-inflammatory, and healing activities induced by the fraction.

Round 2

Reviewer 1 Report

The manuscript describes data about one plant extract, which was subjected to simple cytotoxicity tests. In addition, authors completed an anti-inflammatory test in vivo as local treatment. It remains unclear why and how this test should add valuable data about the therapeutic antileukemic activity. The only interesting finding is that the cytotoxic efficacy remains similar in HL-60 and K-562 cells. The latter one is known to express high amounts of BCR-ABL. The experimental points do not fit well to the dose-response curve in K-562 cells. Fig. 1 contains column diagram, which is not necessary.

Most important disadvantages and weaknesses of the study are:

1. Lack of tests for the cell death type induced such as caspase activation, nuclear morphology, induction of autophagy etc.

2. No positive control compound (antileukemic drug in use) was included.

3. None of the identified compound was tested as pure compound in order to support the antileukemic activity estimated.

Author Response

Comment: The manuscript describes data about one plant extract, which was subjected to simple cytotoxicity tests. In addition, authors completed an anti-inflammatory test in vivo as local treatment. It remains unclear why and how this test should add valuable data about the therapeutic antileukemic activity. The only interesting finding is that the cytotoxic efficacy remains similar in HL-60 and K-562 cells. The latter one is known to express high amounts of BCR-ABL. The experimental points do not fit well to the dose-response curve in K-562 cells. Fig. 1 contains column diagram, which is not necessary.

Response: The aim of this study was to demonstrate the pharmacological and nutraceutical potential of the Polyphenol-Enriched Fraction Extracted From Leaves of Myrtus Communis L. (PEMC). Our data show that PEMC has a broad range of beneficial activities, which could be explored in the formulation of bioactive foods and nutraceuticals. Thus, the present study is within the aim and scope of the special issue “Mediterranean diet: nutraceuticals; metabolic syndrome; environment; neuroendocrine-immune system”.

Based on our results, prospective studies using compounds isolated from this extract may be conducted to further characterize their pharmaceutical activities. Regarding the fitting of experimental points to the dose-response curve for the K-562 cells. Indeed, the points did not fit well the curve. We carefully re-examined and reviewed all data and carried out the analysis again, adjusting for outliers using a robust form of non-linear regression. After the adjustment, the points became a bit closer, but there are still a certain deviation from the curve. Nevertheless, the acquired R-squared value (R2: 0.8696; R2>85%; Sy.x: 15.18) shows a reasonable goodness-of-fit, which enables to consider the strength of the relationship between the two variables analyzed (the independent and dependent ones) as being relatively strong. The column diagram is shown in the figure, since it clearly demonstrates the ability of PEMC to kill cancer cells depending on the concentration applied. The column diagram is also important to show the significant difference in cell viability observed between treated and untreated cells (Dunnet’s test).

  1. Lack of tests for the cell death type induced such as caspase activation, nuclear morphology, induction of autophagy etc.

Response: First, the objective of the the present investigation was to find the chemical composition of the PEMC and to demonstrate its pharmaceutical potential to prevent and treat noncommunicable diseases, such as leukemia.

In future, we intend to conduct experiments to further characterize the mechanism of action of PEMC using isolated constituents of the fraction. 

  1. No positive control compound (antileukemic drug in use) was included.

Response: We did not include an antileukemic drug in the panning of experiments, since the main objective of the present study was to show the anticancer potential of the PEMC, and not to propose an alternative new drug or an alternative treatment of the disease. Therefore, there is no need to add an anticancer drug as positive control for this purpose. We did use a positive control in the present study, it was the DMSO.

As mentioned above, in future, we are going to conduct in vitro and in vivo studies using isolated compounds. For that future investigation we plan to use a commercially available anticancer drug as positive control.

  1. None of the identified compound was tested as pure compound in order to support the antileukemic activity estimated.

Response: As mentioned earlier, in this study we characterize different biological activities of PEMC by showing its pharmaceutical and nutraceutical potential. In future, we have the intention to isolate the compounds identified in this PEMC and to perform more detailed experiments .

Reviewer 3 Report

I believe the manuscript has been sufficiently improved.

Author Response

We thank the Reviewer for his/her comments and time dedicated to the revision of the manuscript.